

# Buffalo-bur (*Solanum rostratum* Dunal) invasiveness, bioactivities, and utilization: a review

Sandra Amarachi Ozuzu[1,2], Rizvi Syed Arif Hussain[1,2], Nigora Kuchkarova[1,2], Gift Donu Fidelis[2], Shixing Zhou[1,2], Théogène Habumugisha[2] and Hua Shao[1,2,3]

[1] State Key Laboratory of Desert and Oasis Ecology, Xinjiang Institute of Ecology and Geography, Chinese Academy of Sciences, Urumqi, China
[2] University of Chinese Academy of Sciences, Beijing, China
[3] Research Center for Ecology and Environment of Central Asia, Xinjiang Institute of Ecology and Geography, Chinese Academy of Sciences, Urumqi, China

## ABSTRACT

*Solanum rostratum* Dunal, belongs to the Solanaceae family and has drawn attention for its intricate interplay of invasiveness, phytochemical composition, and potential bioactivities. Notably invasive, *S. rostratum* employs adaptive mechanisms during senescence, featuring thorn formation on leaves, fruits, and stems seed self-propulsion, and resistance to drought. This adaptability has led to its proliferation in countries such as China, Canada, and Australia, extending beyond its Mexican origin. Despite its invasive historical reputation, recent studies unveil a rich array of phytochemicals in *S. rostratum*, suggesting untapped economic potential due to under-exploration. This review delves into exploring the potential uses of *S. rostratum* while elucidating the bioactive compounds associated with diverse identified bioactivities. In terms of phytochemistry, *S. rostratum* reveals an abundance of various bioactive compounds, including alkaloids, flavonoids, phenols, saponins, and glycosides. These compounds confer a range of beneficial bioactivities, encompassing antioxidant, antifungal, anti-carcinogenic, anti-inflammatory, phytotoxic, and pesticidal properties. This positions *S. rostratum* as a reservoir of valuable chemical constituents with potential applications, particularly in medicine and agriculture. The review provides comprehensive insights into the phytochemistry, bioactivities, and bioactivity-guided fractionation of *S. rostratum*. In this review, we focus on the potential utilization of *S. rostratum* by emphasizing its phytochemical profile, which holds promise for diverse applications. This review is the first that advocates for further exploration and research to unlock the plant's full potential for both economic and environmental benefit.

## INTRODUCTION

Phytochemicals are secondary metabolites specifically synthesized by plants, exerting substantial effects on human health (*Arora & Suvartan, 2020*). Additionally, they play a crucial role in safeguarding plants against diseases and contribute significantly to the development of color, flavor, and fragrance in plants (*Koche, Shirsat & Kawale, 2016*).

Corresponding author
Hua Shao, shaohua@ms.xjb.ac.cn

Phytochemicals exhibiting biological activities (bioactivities) are termed "biologically active compounds". Over the years, numerous studies have been conducted to investigate the significance of various phytochemicals in plants. For instance, research has underscored the economic value of purple maize, attributing it to its abundance in flavonoids, phenolic compounds, and anthocyanin. This plant serves essential roles in the health sector, functioning as an anti-mutagenic, antioxidant, anti-angiogenic, anti-inflammatory, and anti-carcinogenic agent (*Jayaprakash et al., 2023*). Additionally, Mexican oregano essential oils contain a wealth of biologically active compounds with diverse properties, including antioxidant, anti-inflammatory, antifungal, ultraviolet defense, anti-glycemic, and cytotoxic activities (*Bautista-Hernández et al., 2021*). In essence, the presence of bioactive phytochemicals is integral to both human and plant health, as they are associated with significant activities that contribute to overall well-being.

The *Solanum* genus, a large genus within the Solanaceae family, comprises approximately 2,800 species (*Wiart, 2006*). It has been a subject of significant interest in both chemical and biological studies over a long period. Numerous plants belonging to the *Solanum* genus have exhibited the presence of diverse phytochemicals responsible for various bioactivities. For instance, *Solanum lycopersicum* is abundant in phenolic compounds, flavonoids, carotenoids, and nucleosides, rendering it valuable as nutraceuticals and anti-mutagenic plants (*Alam et al., 2019*) The fruits of *Solanum anguivi* are suggested to possess antidiabetic properties, potentially attributed to a range of phytochemicals, including saponins, phenolics, alkaloids, ascorbic acid, and flavonoids (*Nakitto et al., 2021*). Similarly, the therapeutic activities observed in the fruits of *Solanum torvum* are attributed to their abundance in alkaloids, flavonoids, phenols, tocopherols, tannins, saponins, and glycosides (*Darkwah et al., 2020*). Lastly, the polyphenol-rich plant *Solanum nigrum* has been shown to alleviate oxidative stress in the liver by reducing bilirubin and liver enzyme levels (*Alam et al., 2022*). In essence, the nutritional value of plants within the *Solanum* genus is enriched by the presence of these phytochemicals. Basically, the nutritional value of plants within the *Solanum* genus is enhanced by the presence of these phytochemicals.

*Solanum rostratum* Dunal, hereafter referred to as *S. rostratum*, holds significance within the *Solanum* genus where it is recognized for its association with invasion, environmental degradation, and various diseases along with its abundance of phytochemicals that demonstrate potential significance (*Wang et al., 2011*; *Zhao et al., 2013*; *Mahklouf, 2016*; *Matzrafi et al., 2023*). *S. rostratum* has been documented in various countries, initially originating from Mexico and certain regions of the USA (Arkansas, Colorado, Idaho, Iowa, Kansas, Minnesota, Montana, Nebraska, New Mexico, North and South Dakota, Oregon, *etc.*), and subsequently spreading to Europe, Asia, Australia, and few African countries, as illustrated in Table 1 and Figs. 1 & 2. Furthermore, there are reports of this plant not only being naturalized in certain areas of South Africa but also exhibiting invasiveness in countries such as China, Canada, Hungary, Libya, and several others, including Russia, Australia, India, and Japan, where the full extent of its invasiveness has yet to be documented (Fig. 3) (*Datiles, 2014*). This sporadic weed grows in open sites such as trenches, roadsides, landfills, and overgrazed farmlands. This weed is also found

growing within crop rows of watermelon, chickpea, onion, corn, and tomato where the presence of thorns poses challenges in its control, because these crops primarily rely on hand weeding as an effective weed management method (*Abu-Nassar & Matzrafi, 2021*). In order to increase its invasiveness, *S. rostratum* develops adaptive mechanisms during senescence such as formation of thorns on the entire plant including its leaves, fruits, and stems (*Long et al., 2019*), seed self-propulsion (*Whalen, 1979*), drought resistant (*Yu et al., 2021*) and release of biochemical constituents particularly root exudates and mycotoxins during allelopathy (*Shao et al., 2017*; *Shao et al., 2022*). This plant has also been reported to displace native plant species by its ability to out-compete them particularly through their aggressive growth ability and competitive nature. A study conducted to assess the growth and competitive ability of *S. rostratum* compared to two coexisting native plants, *Leymus chinensis* and *Agropyron cristatum*, in China reported that *S. rostratum* showed remarkable growth over the native species by demonstrating higher aboveground and total biomass as well as competitive advantage under both favorable and non-favorable Nitrogen treatments (*Sun et al., 2023*). Also, it has been reported that *S. rostratum* can serve as a host for viruses such as the tomato brown rugose fruit virus (a RNA-based virus) and potexviruses, which have been reported to infect tomatoes thereby causing a reduction in tomato production, and consequently resulting in its loss (*Matzrafi et al., 2023*). Another report showed that some soil fungi isolated from the rhizosphere of *S. rostratum* has the ability to cause negative impact on the growth of some common grassland plants like *Poa pratensis* and *Amaranthus retroflexus* by causing strong growth inhibition on these plants (*Shi et al., 2022*). The potential uses of *S. rostratum* have not been fully explored due to the plant being widely considered invasive.

Despite the prevalent negative perception surrounding this plant, recent studies report that *S. rostratum* is abundant in various phytochemicals, demonstrating some potential beneficial bioactivities and rendering it economically valuable (*Huang et al., 2017*; *Omar et al., 2018*; *Liu et al., 2021*; *Shixing et al., 2021*). To suggest potential avenues for further exploration, a comprehensive literature review was conducted to evaluate the phytochemistry and bioactivities of *S. rostratum*. This review serves as a foundation for future research, aiming to facilitate an optimal exploration of the inherent potential within *S. rostratum*.

## Review methodology

Our comprehensive review involved an exhaustive search across various reputable scientific platforms and journal databases to gather a diverse array of research papers. The primary sources included Web of Science, Wiley Online Library, Cambridge University Press, ScienceDirect, PubMed Centre, Google Scholar, and ACS Publications. Among these, the Web of Science emerged as the predominant platform, contributing significantly to the wealth of literature examined. To optimize our search strategy, we strategically employed a range of keywords, including "*Solanum*", "*S. rostratum*", "*S. rostratum* + distribution", "*S. rostratum* + invasive", "*S. rostratum* + purification", and "*S. rostratum* + phytochemistry". These keywords were meticulously chosen to encompass diverse aspects of our research objectives, ensuring a comprehensive exploration of the literature

**Table 1** The distribution table of *S. rostratum* Dunal.

| Continent | Country | Origin | Author |
|---|---|---|---|
| Africa | Libya | Introduced | *Mahklouf (2016)* and *Chelghoum et al. (2020)* |
| | South Africa | Introduced | *Parsons & Cuthbertson (2001)* |
| | Morocco and Tunisia | Introduced | *Royal Botanic Gardens Kew (1813)* |
| | Algeria | Introduced | *Chelghoum et al. (2020)* |
| Asia | Azerbaijan, Kazakhstan, and Bangladesh | Introduced | *EPPO (2014)* |
| | Uzbekistan | Introduced | *Royal Botanic Gardens Kew (1813)* |
| | China-Inner Mongolia, Beijing, Heibei, Jilin, Liaoning, Shanxi, and Xinjiang | Introduced | *Zhao et al. (2013)* and *Yu et al. (2021)* |
| | Taiwan | Introduced | *Taiwan Plant Names (2014)* |
| | South Korea | Introduced | *Cho & Kim (1997)* and *Seebens et al. (2017)* |
| | India | Introduced | *Som (1976)* |
| | Japan | Introduced | *Randall (2012)* |
| Australia | Australia | Introduced | *Seebens et al. (2017)* and *Royal Botanic Gardens Kew (1813)* |
| Europe | Denmark and Hungary | Introduced | *Randall (2012)* |
| | France and Greece | Introduced | *Royal Botanic Gardens Kew (1813)* |
| | Germany, Bulgaria, Moldova, and Slovakia | Introduced | *EPPO (2014)* |
| | Russia and Ukraine | | *EPPO (2014)* |
| | United Kingdom | Introduced | *Vallejo-Marin (2014)* |
| | Portugal, Spain, Turkey, Israel, Ireland, Sweden, Italy, Croatia, Romania, and Belarus | Introduced | *GBIF.org (2023)* |
| | Lithuania, Latvia, Belgium, and Norway | Introduced | *Seebens et al. (2017)* |
| North America | Mexico | Native | *Whalen (1979)* |
| | United States-Virgin Island | Introduced | *Acevedo-Rodríguez & Strong (2012)* |
| | United States-Arkansas, Colorado, Idaho, Iowa, Kansas, Minnesota, Montana, Nebraska, New Mexico, Oregon, North and South Dakota | Native | *USDA-ARS (2014)* |
| | United States-Arizona, California, Florida, Georgia, Kentucky, Maryland, Mexico Central, Mexico Gulf, Mexico Northeast, Mexico Northwest, Mexico Southeast, Mexico Southwest, Mississippi, Tennessee, Utah, and West Virginia | Native | *Royal Botanic Gardens Kew (1813)* |
| | United States-Washington, Oklahoma, and Wyoming | Introduced | *Randall (2012)* |
| | United States-Missouri | Introduced | *Whalen (1979)* |
| | United States-Texas | Native | *Whalen (1979)* |
| | Canada | Introduced | *Seebens et al. (2017)* |
| Oceania | New Zealand | Introduced | *Randall (2012)* |

landscape related to *S. rostratum*. Subsequently, the identified papers underwent a rigorous selection process, with only the most relevant and germane studies being included and cited in our analysis. This meticulous curation aimed to uphold the quality and precision of our review, aligning with the specific focus and objectives of our research.

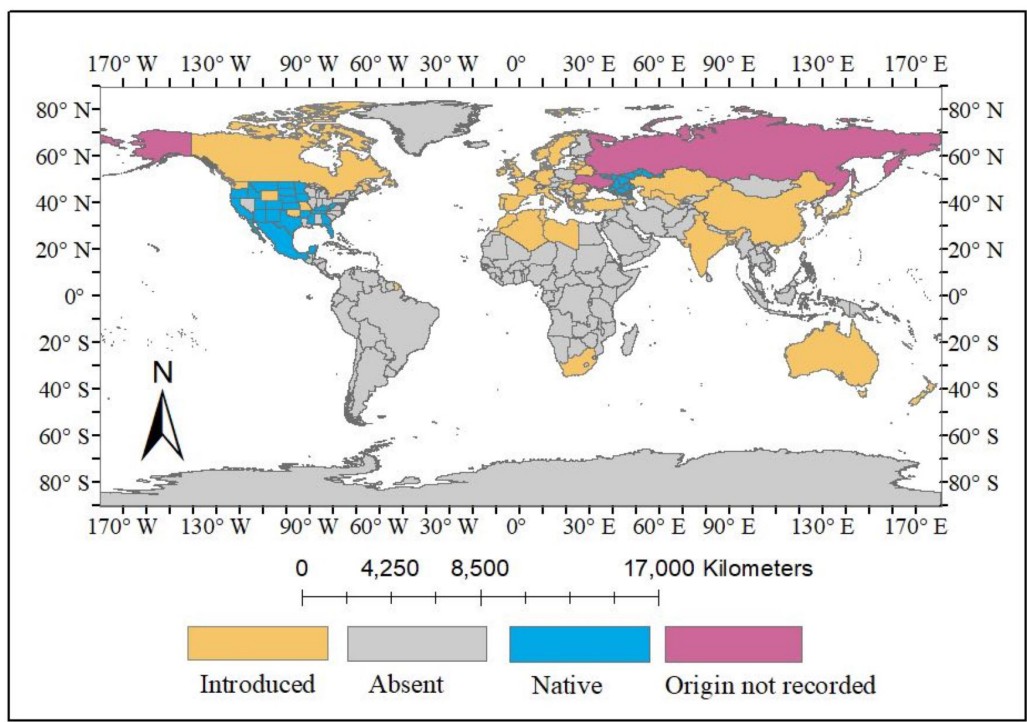

**Figure 1 The origin of *S. rostratum* Dunal.** Changes made on the map primarily indicate the origin of *S. rostratum* Dunal. Map source credit: *Runfola et al. (2020)* (http://www.geoboundaries.org).

## Phytochemical constituents present in *s. rostratum*
### Alkaloids

Alkaloids represent a diverse group of secondary chemical compounds widely present in plants, recognized as a defense mechanism against insects, microorganisms, animals, and humans due to their structural variety. In *S. rostratum*, the main alkaloids (Fig. 4) include pyrrole alkaloids, methylprotodioscin, phenylethylamine, solanine, chaconine, and glycosidic alkaloids (*Novruzov, Aslanov & Ismailov, 1973*; *Bah et al., 2004*; *Zhang et al., 2012*; *Liu et al., 2019*; *Liu et al., 2021*). Pyrrole alkaloids are a class of naturally occurring compounds that contain a five-membered ring composed of four carbon atoms and one nitrogen atom found in various plants, fungi, and marine organisms where it serves as a chemical defense against predators and protection against environmental stresses (*Singh & Majik, 2019*). *Liu et al. (2021)* recently identified and isolated three pairs of novel enantiomeric pyrrole alkaloids—(2′ R)-Caffeicpyrrole A and (2′ S)-Caffeicpyrrole A, (2′ R)-Caffeicpyrrole B, and (2′ S)-Caffeicpyrrole B, and (2′ R)-Caffeicpyrrole C and (2′ S)-Caffeicpyrrole C—from the leaves of *S. rostratum* where it confirms its efficacy as an antifeedant and growth-inhibitory agent against *Henosepilachna vigintioctomaculata,* an invasive herbivore of *S. rostratum*.

Furthermore, other studies indicated that phenylethylamine, extracted from the aerial parts of *S. rostratum*, displayed anti-feedant activity against *Helicoverpa armigera* (*Liu et al., 2019*). Also, methylprotodioscin, isolated from the aerial parts of *S. rostratum*,
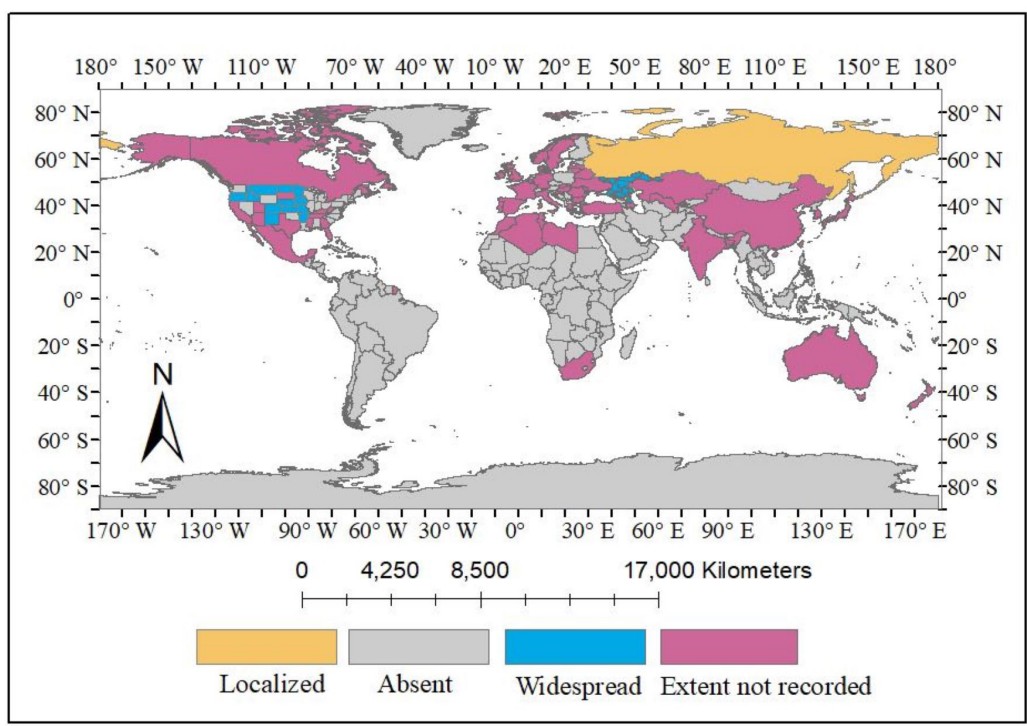

**Figure 2 The global distribution of *S. rostratum* Dunal.** Changes made on the map primarily indicate the global distribution of *S. rostratum* Dunal. Map source credit: *Runfola et al. (2020)* (http://www.geoboundaries.org).

demonstrated its capability to induce cytotoxic effects on human cancer cells (*Bah et al., 2004*). In another study, solanine and chaconine were isolated and identified from various parts of *S. rostratum,* demonstrating their potential in controlling pests, especially those with chewing and sucking mouthparts on a variety of vegetable crops (*Zhang et al., 2012*).

### Flavonoids

*S. rostratum*, classified in the Androceras section series, is recognized for its abundance of flavonoids (*Huang et al., 2017*). Important flavonoids have been isolated and identified from *S. rostratum* (Fig. 5). Quercetin is one of the major flavonoids primarily sourced from the aerial part of *S. rostratum* that plays a crucial role in neutralizing free radicals in the body, promotes seed germination and plant development in both stressful and non-stressful environments (*Huang et al., 2017*; *Omar et al., 2018*). As a plant-derived aglycone, quercetin has demonstrated superior disease-fighting abilities compared to curcumin, exhibiting higher reduction potential, a 3.5-fold increase in total antioxidant capacity, and a reduction in reactive oxygen species (ROS) and nitric oxide (NO) in LPS-stimulated human THP-1 acute monocytic leukemia cells (*Zhang et al., 2011*). Additionally, quercetin mitigates mycotoxin-induced endoplasmic reticulum stress and apoptosis, providing cellular protection (*Yang et al., 2020*). Another aglycone flavonoid, kaempferol, has been successfully isolated from *S. rostratum* (*Ibarra-Alvarado et al., 2010*; *Huang et al., 2017*).

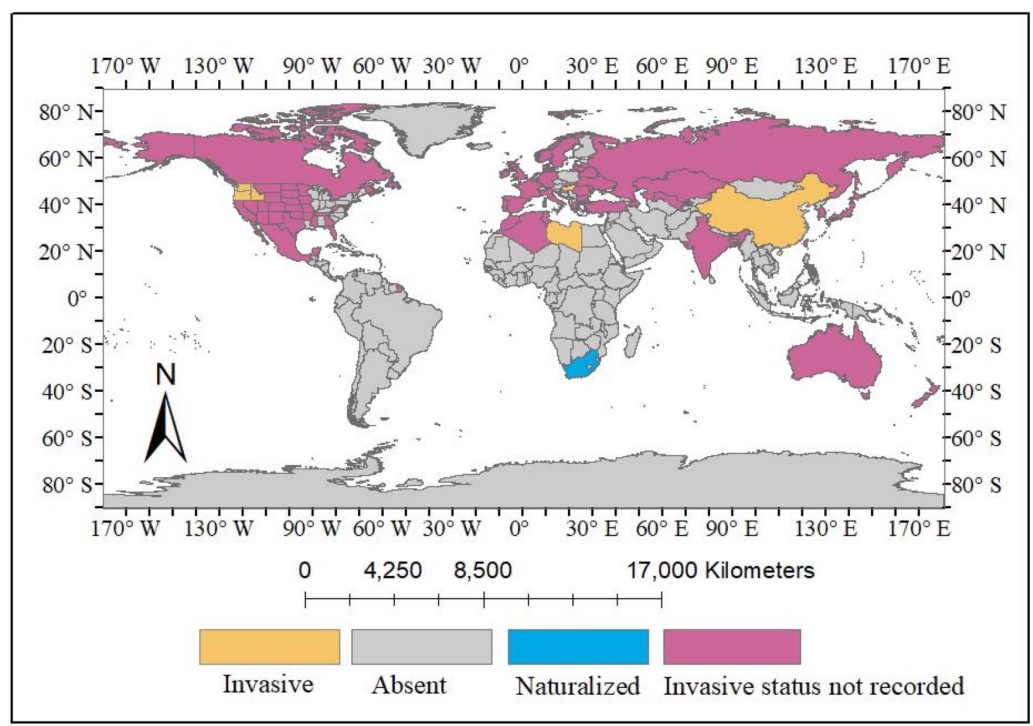

**Figure 3** **The global invasiveness of *S. rostratum* Dunal.** Changes made on the map primarily indicate the global invasiveness of *S. rostratum* Dunal. Map source credit: *Runfola et al. (2020)* (http://www.geoboundaries.orgs).

**Figure 4** **Chemical structure of important alkaloids isolated and identified from *S. rostratum*.** (A) Pyrrole alkloids. (B) Methylprotodioscin. (C) Phenylethylamine. (D) Chaconine. (E) Solanine.

**Figure 5 Chemical structure of important flavonoids isolated and identified from *S. rostratum*.** (A) Quercetin. (B) Kaempferol. (C) Hesperitin. (D) Luteolin. (E) Rutin. (F) Isorhamnetin-3-O-glucoside.

Kaempferol demonstrates potential in treating obesity-related ailments by promoting lipolysis and suppressing adipogenesis in 3T3-L1 adipocytes, a mouse pre-adipocyte cell line (*Muni Swamy et al., 2020*). It has also been reported to inhibit the growth of pancreatic cancer cells through apoptosis induction (*Zhang et al., 2008*). Hesperitin, another major flavonoid found in *S. rostratum* (*Ibarra-Alvarado et al., 2010*) has been associated with reducing cancer cell proliferation by inducing cell cycle arrest and exhibiting cytostatic effects (*De Oliveira, Santos & Fernandes, 2019*). Likewise, luteolin, a flavonoid in *S. rostratum* (*Wang et al., 2015*) has been linked to inhibiting tumor growth by influencing cellular processes such as apoptosis, angiogenesis, migration, and cell cycle progression (*Çetinkaya & Baran, 2023*) as well as suppressing some proinflammatory mediators such as IL-6, IL-8, TNF-$\alpha$, and COX-2 which in turn regulates different skin inflammatory processes (*Gendrisch et al., 2021*).

## Other compounds

The phenolic constituents in *S. rostratum* contributes to its potential economic significance where the total phenols identified in *S. rostratum* (*Valadez Vega et al., 2019*) have been reported to have the ability to mitigate oxidative stress (*Phuyal et al., 2020*), function as food preservatives (*Derakhshan et al., 2018*), and exhibit substantial significance ($p < 0.05$) in the treatment of callus using elicitors (*Ali, El-Nour & Yagi, 2018*). Terpene glycosides are non-volatile natural products responsible for a number of bioactivities in plants (*Schwab, Fischer & Wüst, 2015*). Linalyl-$\beta$-glucopyranoside, a terpene glycoside, was found to be present in the chloroform extract from the aerial part of *S. rostratum* after analysis with the Folin–Ciocalteu reagent (*Omar et al., 2018*). Linalyl-$\beta$-glucopyranoside also referred to as linalool glucoside, functions as an aroma precursor subsequent to the release of linalool

**Figure 6 Chemical structure of other compounds isolated from *S. rostratum*.** (A) Apigenin-7-O-glucoside. (B) Linalyl- $\beta$-glucopyranoside.

(*Moon et al., 1994*). The chemical structure of linalyl-$\beta$-glucopyranoside is illustrated in Fig. 6.

Furthermore, a diverse array of phytochemicals present in *S. rostratum* contributing to distinct bioactivities have been compiled in Table 2.

## Biological activities associated with *S. rostratum*
### Antioxidant activity

An antioxidant is a substance that inhibits or reduces oxidative damage caused by free radicals in the body that are associated with cell damage, aging, and chronic diseases. Free radicals are highly reactive molecules or ions that can damage cells and contribute to various health issues, including aging and chronic diseases specifically in male. *S. rostratum* are good anti-oxidants because of their richness in flavonoids like 5,7,8, 4′-tetrahydrox y-3-m ethoxyflavon e-8-O- $\beta$-d-xylopyranoside, querceti n-3-O-$\beta$-D-g alactopyranoside, quercetin, quercetin 3′-methox y-3-O-$\beta$-D-g alactopyranoside and kaempfero l-3-O-$\beta$-D-g lucopyranoside (*Huang et al., 2017*) having the potential to be used as promoters for insulin secretion and enhancement of insulin resistance in man (*Yi et al., 2021*). Notably, quercetin, a flavonoid in *S. rostratum*, has demonstrated significant inhibitory activity against the stable free radical compound 2,2-diphenyl-1-picrylhydrazyl(DPPH), surpassing the effectiveness of ascorbic acid (*Azeem et al., 2023*). *Omar et al. (2018)*, in their examination of phytochemical constituents in *S. rostratum*, also investigated its antioxidant activity where the result revealed the chloroform extract of the plant exhibited an exceptional radical scavenging ability with an $IC_{50}$ value of 0066 $\pm$ 0.001 mg/ml. Additionally, the isolation of isorhamnetin-3-O-glucoside, a flavonoid glycoside, has been reported to possess the potential to treat cardiovascular-related diseases by regulating specific signaling pathways such as phosphatidylinositol 3-kinase/ protein kinase B(PI3K/AKT/PKB) signaling pathways, nuclear factor kappa-light-chain-enhancer

**Table 2** Bioactive Compounds Present in *S. rostratum* Dunal.

| Bioactive chemical | Bioactive compound | Plant part | Bioactivities | Separation Techniques | Author |
|---|---|---|---|---|---|
| | Pyrrole Alkaloids | Leaves | Anti-feedants, growth inhibitory activities in insect like the *Henosepilachna vigintioctomaculata*. | CC, LC & HPLC | *Zhixiang et al. (2021)* |
| | Methylprotodioscin | Flowers | Anti-inflammatory | CC | *Baytelman (1993)* and *Bah et al. (2004)* |
| | Phenylethylamine A | Aerial Part | Anti-feedants: activity against *Helicoverpa armigera* | HPLC & GC | *Liu et al. (2019)* |
| Alkaloids | $\alpha$-solanine, $\beta$-solanine, $\gamma$-solanine $\alpha$-chaconine, $\beta$-chaconine, and $\gamma$-chaconine | Entirety of Plant | Insecticides and Anti-feedants: activities against lepidopterous larva and aphides. | Filtration and Distillation | *Zhang et al. (2012)* |
| | Glycosidic alkaloids | Aerial Part | Anti-inflammatory, antimicrobial, and anti-carcinogenic activities | TLC | *Novruzov, Aslanov & Ismailov (1973)* and *Abu Bakar Siddique & Brunton (2019)* |
| Flavonoids | 5,7,8, 4′-tetrahydroxy-3-m ethoxyflavone-8-O-$\beta$-d-xylopyranoside, querceti n-3-O-$\beta$-D-g alactopyranoside, quercetin, quercetin 3′-methox y-3-O-$\beta$-D-g alactopyranoside and kaempfero l-3-O-$\beta$-D-g lucopyranoside | Aerial Part | Antioxidants | CC & TLC | *Huang et al. (2017)* |
| | Flavonoid glycosides; isorhamnetin-3-O-glucoside. | Aerial Part | Antioxidants | SGC, TLC & HPLC | *Omar et al. (2018)* |
| | Hyperoside, Astragalin, 3′-O-methylquercetin 3-O-$\beta$-D-galactopyranoside, and 3′-O-methylquercetin 3-O-$\beta$-D-glucopyranoside | Entirety of Plant | Antioxidants | HPLC & HSCC | *Shao et al. (2018)* |
| | Luteolin | Aerial Part | Antioxidants and anti-inflammatory | LS & MS. | *Wang et al. (2015)* and *Shukla et al. (2019)* |
| | Hesperitin, quercetin, catechin, kaempferol, and rutin | Entirety of Plant | Antioxidants | HPLC | *Ibarra-Alvarado et al. (2010)* |

**Table 2** (*continued*)

| Bioactive chemical | Bioactive compound | Plant part | Bioactivities | Separation Techniques | Author |
|---|---|---|---|---|---|
| Phenolic Compounds | Total phenols | Leaves | Antioxidants | Spectrophotometry | *Valadez Vega et al. (2019)* |
| | Apigenin-7-O-glucoside , | Aerial Part | Antioxidants | SGC, TLC & HPLC | *Omar et al. (2018)* |
| Terpenes | Terpene glycosides; Linalyl- $\beta$-glucopyranoside | Aerial Part | Antioxidants | SGC, TLC & HPLC | *Omar et al. (2018)* |

**Notes.**

TLC, thin-layer chromatography; CC, column chromatography; LC, liquid chromatography; HPLC, high performance liquid chromatography; GC, gel chromatography; SGC, silica-gel chromatography; HSCC, high speed countercurrent chromatography; MS, mass spectrophotometry.

of activated B cells (NF-$\kappa$B) transcription factor protein, mitogen-activated protein kinase (MAPK) and the expression of associated kinases and cytokines (*Omar et al., 2018*).

In animals like rat, *S. rostratum* has been observed to have the ability to induce concentration-dependent contraction of rat aortic rings, suggesting potential health benefits for treating conditions such as venous insufficiency (*Ibarra-Alvarado et al., 2010*).

## Anti-inflammatory activity

*S. rostratum,* has been recognized for its profound anti-inflammatory activity, making it a compelling subject of study in the realm of natural medicine. The plant's anti-inflammatory effects can be attributed to its rich phytochemical composition, including alkaloids, flavonoids, and phenolic compounds. These bioactive constituents have demonstrated the ability to modulate key inflammatory pathways and mediators. Studies have revealed that *S. rostratum* extracts possess inhibitory effects on pro-inflammatory cytokines, such as tumor necrosis factor-alpha (TNF-$\alpha$) and interleukin-6 (IL-6), crucial players in the inflammatory response. Moreover, the plant's extracts have been found to downregulate the activity of enzymes like cyclooxygenase-2 (COX-2) and inducible nitric oxide synthase (iNOS), further attenuating inflammatory processes. This anti-inflammatory potential positions *S. rostratum* as a valuable resource for traditional medicine and suggests its potential application in the development of anti-inflammatory drugs. The elucidation of the specific mechanisms underlying its anti-inflammatory activity holds promise for the advancement of therapeutic interventions targeting inflammatory disorders. In humans, sitosteryl glucosides have demonstrated *in vivo* anti-inflammatory activity on important pro-inflammatory enzymes such as cyclooxygenases (COX-1 and COX-2 (ovine/human)) and lipoxygenases (human recombinant-5-LOX enzyme) that are mediators for a number of inflammations and sensitized diseases observed in the body (*El-Feky & El-Rashedy, 2023*). Additionally, methylprotodioscin, found in *S. rostratum*, has been reported for its application in treating chronic intestinal inflammation by ameliorating intestinal mucosal inflammation through the regulation of intestinal immunity, thereby enhancing the differentiation of the intestinal barrier (*Zhang et al., 2015*).

In animals, the sitosteryl glucosides compound isolated from *S. rostratum* has also been recognized for its anti-inflammatory properties, particularly in combating Tissue-type Plasminogen Activator (TPA)-induced mouse ear edema (*Alvarez et al., 2009*) as well as Carrageenan induced paw edema (*Naikwadi, Phatangare & Mane, 2022*).
## Anti-carcinogenic activity

*S. rostratum,* has exhibited noteworthy anti-carcinogenic activities, marking it as a promising candidate in cancer research. Extensive studies have uncovered bioactive compounds within *S. rostratum* that contribute to its anti-carcinogenic properties. Alkaloids, flavonoids, and phenols found in this plant have demonstrated inhibitory effects on cancer cell proliferation and metastasis. These compounds, through various molecular mechanisms, showcase potential in suppressing the growth of cancer cells, inducing apoptosis, and impeding angiogenesis. Additionally, the richness of phytochemical composition present in *S. rostratum* has shown promise in mitigating oxidative stress, a factor implicated in cancer development. The plant's anti-inflammatory properties further contribute to its anti-carcinogenic potential, as chronic inflammation is closely linked to cancer progression. The exploration of anti-carcinogenic activities of *S. rostratum* not only underscores its significance in traditional medicine but also presents an avenue for the development of novel therapeutic interventions in the field of oncology. Further research is warranted to elucidate specific molecular pathways involved and optimize the utilization of *S. rostratum* in potential anti-cancer treatments. Due to the abundant phenolic content found in *S. rostratum*, the ethyl acetate extract from this plant has been noted for its ability to exhibit anti-carcinogenic activity against human malignant cell lines, particularly those associated with breast adenocarcinoma and cervix squamous cancer. This was substantiated by a significant decrease in cell viability (*Valadez Vega et al., 2019*). Additionally, the aerial portion of *S. rostratum* is employed in the preparation of infusions, that helps regulate vaginal fluids, disinfect genital areas, and serve as a supplementary element in the treatment of uterine cancer.

## Antifungal and antimicrobial activity

In Mexico, the hydroalcoholic extract of *S. rostratum* has also been noted for its ability to impede the growth of key fungi (isolates 501 and 498 of *Candida albicans*, isolates 434 and 514 of *Aspergillus fumigatus*, isolates 1526 and 1591 of *Histoplasma capsulatum*, and isolates 168 and 167 of *Coccidioides immitis*) associated with pulmonary mycoses, a fungal disease characterized by lung tissue necrosis and the development of vascular thrombosis (*Alanís-Garza et al., 2007*).

Additionally, *S. rostratum* exhibits antifungal activity against *Curvularia lunata*, a causal fungus of corn leaf spot disease, where it causes a significant growth inhibition (inhibition by 80%) when exposed to the extract of this plant (*Hernandez-Rodriguez et al., 2018*).

## Herbicidal/phytotoxic activity

*S. rostratum* demonstrates considerable potential in controlling common weeds that can adversely impact the growth of economically important plants. Extracts from various parts of *S. rostratum* have been found to inhibit the seed germination and development of several common weeds like *Poa annua*, *Amaranthus retroflexus*, and a common vegetable, *Solanum lycopersicum* (*Ping, Zhu & Zhang, 2012*; *Shao, 2015*; *Zhou et al., 2021*). The soil surrounding *S. rostratum* harbors various fungi, including *Aspergillus flavus*, which exhibits allelopathic behavior toward *Amaranthus retroflexus, Thlaspi arvense,* and *Poa annua* by producing

the mycotoxin, kojic acid, that affects the normal functioning of these plants (*Shao et al., 2017*). Furthermore, essential oils extracted from *S. rostratum* have demonstrated efficacy in inhibiting the growth and development of *Amaranthus retroflexus*, impacting shoot and root length growth as well as seed germination. *S. rostratum* also exerts phytotoxic effects on *Arabidopsis thaliana*, a model organism, by inhibiting root elongation and seed germination (*Liu et al., 2023*). Additionally, secondary metabolites such as thymine, adenosine, and cerevisterol, isolated from an endophytic fungus *Purpureocillium* sp. found in *S. rostratum*, have been reported to exhibit inhibitory activity against the growth of *Amaranthus retroflexus*, through the inhibition of seed sprouting and a reduction in root length (*Kuchkarova et al., 2020*).

## Insecticidal and pesticidal activities

*S. rostratum* has demonstrated growth-inhibitory effects against *Helicoverpa armigera*, a pest that primarily feeds on Fabaceae, Malvaceae, Poaceae, and Solanaceae plants (*Cunningham & Zalucki, 2014*), where it has also shown to cause an inhibition in the feeding process of this pest thus positioning it as a viable pesticide (*Liu et al., 2019*). Here, the ethyl acetate fraction demonstrated the best anti-feedant activity by 80% against *Helicoverpa armigera* at the highest concentration (0.5 mg/mL), outperforming water and petroleum ether fractions (*Liu et al., 2019*). In a similar investigation, *S. rostratum* displayed antifeedant and growth-inhibitory activity against *Henosepilachna vigintioctomaculata* through the isolation of three pairs of novel enantiomeric pyrrole alkaloids its leaves. The results revealed enantioselectivity with all enantiomers demonstrating both antifeedant and growth-inhibitory activities having the R configuration at C-2′ exhibiting the most potent effects (*Liu et al., 2021*). Last but not the least, the extracts of *S. rostratum* have been utilized as insecticides to control lepidopterous larvae and aphids on fruit and vegetable crops due to their high toxicity, repellent properties, anti-feeding effects, and stomach poisoning effects, thereby mitigating the impact of these insects on crops and reducing economic losses (*Zhang et al., 2012*).

## CONCLUSION AND FUTURE PERSPECTIVES

Due to phenotypic plasticity (*Yu et al., 2021*) and other previously mentioned adaptive mechanisms, *S. rostratum* can withstand both extreme and mild environmental conditions, consequently resulting in the displacement and reduction of native species as well as changes to the vegetation structure (*Hejda, Pyšek & Jarošík, 2009*). This study unveils a diverse array of phytochemical constituents in *S. rostratum*, showcasing its potential significance in the pharmacological/medical and agricultural sectors. This review highlights the importance of *S. rostratum* as an antioxidant against free radicals in humans which can aid in preventing cell damage and cardiovascular-related diseases, enhancing its pharmacological/medical relevance. In animals, *S. rostratum* contains the notable anti-inflammatory compound sitosteryl glucoside, reported to be effective against TPA-induced mouse ear edema and Carrageenan-induced paw edema, suggesting potential applications in veterinary medicine. Moreover, the abundance of phenolic compounds in *S. rostratum* contributes to its anti-carcinogenic properties against human malignant cell lines, specifically breast

adenocarcinoma, cervix squamous cancer, and uterine cancer. The review also highlights antimicrobial activities against both human pathogens and phytopathogens. Additionally, *S. rostratum* have the potentials to serve as herbicides, insecticides, and pesticides, promoting sustainable agriculture that promotes a healthy environment.

Despite being considered invasive, this study emphasizes the need for future research to fully comprehend appropriate exploration methods, mitigating the impact of under-exploration and paving the way for maximal exploration. Continuous isolation and identification of different phytochemicals with relevant bioactivities are crucial in harnessing *S. rostratum* potentials particularly in the field of medicine (ethnobotany) and agriculture. From the perspectives of ethnopharmacology and phytochemistry, *S. rostratum* remains insufficiently researched in this biome. The study advocates for a focus on maximizing the potential benefits of *S. rostratum* rather than solely concentrating on its invasiveness.

### Funding
This research was funded by the Natural Science Foundation of Xinjiang Uygur Autonomous Region (2022D01D02) and the Alliance of International Science Organizations (ANSO) Scholarships for Young Talents University of Chinese Academy of Sciences, and the Third Xinjiang Scientific Expedition Program (2022xjkk1505). The funders had no role in study design, data collection and analysis, decision to publish, or preparation of the manuscript.

### Grant Disclosures
The following grant information was disclosed by the authors:
Natural Science Foundation of Xinjiang Uygur Autonomous Region: 2022D01D02.
Alliance of International Science Organizations (ANSO) Scholarships for Young Talents University of Chinese Academy of Sciences.
Third Xinjiang Scientific Expedition Program: 2022xjkk1505.

### Competing Interests
The authors declare there are no competing interests.

### Author Contributions
- Sandra Amarachi Ozuzu conceived and designed the experiments, performed the experiments, analyzed the data, prepared figures and/or tables, authored or reviewed drafts of the article, and approved the final draft.
- Rizvi Syed Arif Hussain performed the experiments, analyzed the data, authored or reviewed drafts of the article, and approved the final draft.
- Nigora Kuchkarova performed the experiments, analyzed the data, authored or reviewed drafts of the article, and approved the final draft.

- Gift Donu Fidelis performed the experiments, analyzed the data, prepared figures and/or tables, authored or reviewed drafts of the article, prepared maps, and approved the final draft.
- Shixing Zhou performed the experiments, analyzed the data, authored or reviewed drafts of the article, and approved the final draft.
- Théogène Habumugisha performed the experiments, analyzed the data, authored or reviewed drafts of the article, and approved the final draft.
- Hua Shao conceived and designed the experiments, performed the experiments, analyzed the data, prepared figures and/or tables, authored or reviewed drafts of the article, and approved the final draft.

### Data Availability
   This is a literature review.

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
