# Peer review of "Buffalo-bur (Solanum rostratum Dunal) invasiveness, bioactivities, and utilization: a review"

_PeerJ, doi:10.7717/peerj.17112_

## Round 0.1 · original submission · Major Revisions

Experts in the field have carefully reviewed your manuscript and have raised a number of serious concerns that preclude its acceptance in the present form. This paper has been recommended for MAJOR REVISIONS. I invite you to respond to the reviewers' comments and revise your paper accordingly. The revised version will be re-evaluated by the original reviewers or in some circumstances by new reviewers. Please be aware that this invitation to revise your manuscript does not guarantee eventual acceptance of your manuscript.

Moreover, thorough English editing is required. Please revise the manuscript taking help from a colleague who is proficient in English and familiar with the subject matter, who can review your manuscript, or contact a professional editing service to review your manuscript. Revise and resubmit accordingly.

**Language Note:** The Academic Editor has identified that the English language must be improved. PeerJ can provide language editing services - please contact us at [email protected] for pricing (be sure to provide your manuscript number and title). Alternatively, you should make your own arrangements to improve the language quality and provide details in your response letter. – PeerJ Staff

·

Basic reporting

In my opinion, this review should be more focused and short.

In the abstract, author state they examine phytochemistry, biological activities invasiveness, distribution, and utilization of this plant. Then, they state at the end of the introduction: "Rather than focusing on the "invasive" qualities of S. rostratum, this review sheds light on its potential uses. To suggest possible options for exploring Buffalo-bur, we reviewed a number of literature that evaluated its phytochemistry and biological activities. We provide a framework for future research that will ensure optimal exploration of S. rostratum potential, regardless of the species' invasive nature". The first half of this manuscript described botanical, biological, invasion and distribution of S. rostratum, while only at page 13 authors start to review phytochemical properties. I have not seen any specific mention for "framework for future research that will ensure optimal exploration of S. rostratum potential", what do you suggests and how? This should be further discussed in a specific section of the review; I suggest "Future Perspectives".

The introduction should include a short botanical, biological and distribution description of S. rostratum. The whole section on species description is redundant and do not serve the focus of the review. Authors should stay focused on the topic to make this review more coherent. I have not reviewed this part since I think it should be removed from the manuscript.

Authors may also choose to present the different activities according to the chemical structure; flavonoids, alkaloids, etc. or their general activity; herbivore, human medicine, etc. However, the way it is written now, there seem to be no specific order which makes it difficult for the reader to follow.

In several places across the text, authors mention different activities of compounds found in S. rostratum. However, specific experiments done with extracts from this plant are not fully specified. Authors should discuss the results of the specific experiments from the research cited. For instance, "In order to evaluate their potential for chemical defense against Henosepilachna vigintioctomaculata, these alkaloids were tested against this invasive herbivore of S. rostratum." What were the results of this test (248-250)? Another example for this, how does phenolic content of buffalo-bur has shown anti-carcinogenic activity (344-346)?

Experimental design

The review methodology is fine. I think that since it is not a meta-analysis, it is not very important to do it in this way.

Validity of the findings

Not relevant.

Additional comments

61-62: S. rostratum have also been found in countries across Europe, Portugal, Spain, Turkey, Israel, etc. These countries were invaded by this species https://www.gbif.org/species/2930922
74-76: This species was also found as potential host of viruses https://journals.plos.org/plosone/article?id=10.1371/journal.pone.0282441
77-79: Authors should add a citation for this sentence.
248-253: When discussing the chemical activity of different compounds found in S. rostratum, authors should distinguish activity against herbivores or human cancer, or according to chemical compounds. Anyway, it should be organized in one order to make it easier for the reader.
321: Here, authors discuss pharmacology, while anti-cancer was already described elsewhere, it should all be combined together.
365-393: This section is redundant, as this is not the focus of the review. Weed management options are important but not this is out of the scope of this review.

Reviewer 2 ·

Basic reporting

.

Experimental design

.

Validity of the findings

.

Additional comments

Attached please find the manual with postils in. Some of the general comments are listed as follow:
1, Shorten the article as some parts as postils suggested and shorten the reference, particularly.
2, Some paragraphs can be merged.
3, Make check of the consistency of the S. rostratum, Solanum rostratum and Buffalo-bur.
4, Polish the conclusion and make it more close to the goal of the article.
Review conclusion: accept after revision.

Annotated reviews are not available for download in order to protect the identity of reviewers who chose to remain anonymous.

Reviewer 3 ·

Basic reporting

I have finished reviewing the manuscript, “Solanum rostratum Dunal Bioactivities, Invasiveness, and Utilization: A review”. The authors investigated the in depth review of phytochemistry and biological activities of Solanum rostratum. However, the manuscript writing is poor. The authors need to do a thorough reading of what they are doing. I have pointed out some major issues in the manuscript.
Title;
In the title add the English name of the plant along with the botanical name
Abstract;
Consider rephrasing "Inadequate exploration has left the potential applications of S. rostratum completely unknown due to its invasiveness." It's a bit vague. You could mention that the potential applications have been underexplored due to the plant's reputation as invasive.

Experimental design

Methodology
• You mention that you selected and cited only the papers most pertinent to your research. Explain the criteria or factors that you used to determine the relevance of the papers. This adds transparency to your methodology.
• Specify the time frame during which you conducted these searches. This is important because it helps readers understand the currency of the literature you've reviewed.
• Overall, your methodology is sound, but providing additional details on your search strategy and criteria for paper selection would enhance the transparency and reliability of your research review.

Validity of the findings

Introduction
• Elaborate on the negative impacts of S. rostratum on native plants, including specific examples or consequences.
• When discussing the spread of S. rostratum to countries like China, Canada, and Australia, consider providing information about the scale and impact of its invasion in these regions.
• While the invasive nature of S. rostratum is mentioned, it would be beneficial to elaborate on its ecological and economic impact. This will provide context for why exploring its potential uses is important.
• The transition from discussing the broader Solanum genus to S. rostratum is abrupt. Consider introducing S. rostratum more smoothly, perhaps by briefly mentioning its relevance in the context of the Solanum genus and its invasive nature. A few studies related to invasive species of Genus Solanum mentioned here (https://doi.org/10.1016/j.ecolind.2023.110053), and (https://doi.org/10.1080/24749508.2023.2179752).

Additional comments

I have finished reviewing the manuscript, “Solanum rostratum Dunal Bioactivities, Invasiveness, and Utilization: A review”. The authors investigated the in depth review of phytochemistry and biological activities of Solanum rostratum. However, the manuscript writing is poor. The authors need to do a thorough reading of what they are doing. I have pointed out some major issues in the manuscript.

---

## Round 0.2 · Minor Revisions

Though the manuscript is significantly improved by the authors, reviewer 1 still has raised some suggestions to improve the manuscript. Please revise considering the comments and resubmit.

·

Basic reporting

Authors have corrected the manuscript according to my previous comments. I have only two minor comments.

Authors wrote: "This sporadic weed grows in open sites like trenches, roadsides, landfills, and overgrazed farmlands."
Abu-Nassar and Matzrafi ( https://doi.org/ 10.3390/plants10020284) showed that this weed can be found also within crop rows of onion, chickpea, corn and tomato.

In the acknowledgment section: "I would like to...", "we" would be more appropriate.

Experimental design

OK

Validity of the findings

OK

Reviewer 3 ·

Basic reporting

Improved

Experimental design

Improved

Validity of the findings

clear

Additional comments

Now the manuscript can be considered for publication

---

## Round 0.3 · accepted · Accept

The manuscript has been significantly improved by the authors and now can be accepted in its current form.